# Update on frequency decline of Northeast Pacific blue whale (*Balaenoptera musculus*) calls

Ally Rice[1]*, Ana Širović[2¤a], John A. Hildebrand[1], Megan Wood[2], Alex Carbaugh-Rutland[2¤b], Simone Baumann-Pickering[1]

**1** Scripps Institution of Oceanography, University of California San Diego, La Jolla, CA, United States of America, **2** Texas A&M University at Galveston, Galveston, TX, United States of America

¤a Current address: Trondhjem Biological Station, Norwegian University of Science and Technology, Trondheim, Norway
¤b Current address: Abess Center for Ecosystem Science and Policy, University of Miami, Coral Gables, FL, United States of America
* arice@ucsd.edu

**Data Availability Statement:** All relevant data are within the paper and its Supporting Information files.

## Abstract

Worldwide, the frequency (pitch) of blue whale (*Balaenoptera musculus*) calls has been decreasing since first recorded in the 1960s. This frequency decline occurs over annual and inter-annual timescales and has recently been documented in other baleen whale species, yet it remains unexplained. In the Northeast Pacific, blue whales produce two calls, or units, that, when regularly repeated, are referred to as song: A and B calls. In this population, frequency decline has thus far only been examined in B calls. In this work, passive acoustic data collected in the Southern California Bight from 2006 to 2019 were examined to determine if A calls are also declining in frequency and whether the call pulse rate was similarly impacted. Additionally, frequency measurements were made for B calls to determine whether the rate of frequency decline is the same as was calculated when this phenomenon was first reported in 2009. We found that A calls decreased at a rate of 0.32 Hz yr$^{-1}$ during this period and that B calls were still decreasing, albeit at a slower rate (0.27 Hz yr$^{-1}$) than reported previously. The A call pulse rate also declined over the course of the study, at a rate of 0.006 pulses/s yr$^{-1}$. With this updated information, we consider the various theories that have been proposed to explain frequency decline in blue whales. We conclude that no current theory adequately accounts for all aspects of this phenomenon and consider the role that individual perception of song frequency may play. To understand the cause behind call frequency decline, future studies might want to explore the function of these songs and the mechanism for their synchronization. The ubiquitous nature of the frequency shift phenomenon may indicate a consistent level of vocal plasticity and fine auditory processing abilities across baleen whale species.

**Funding:** Funding was provided to JAH by the US Navy Pacific Fleet (https://www.navymarinespeciesmonitoring.us/), grant number N00244-10-C-0021. The funders had no role in the study design, data collection and analysis, decision to publish, or preparation of the manuscript.

**Competing interests:** The authors have declared that no competing interests exist.

## Introduction

The calls produced by multiple blue whale (*Balaenoptera musculus*) populations have been declining in frequency since they were first recorded in the 1960s [1]. This phenomenon was first reported by McDonald *et al.* [1], and subsequent studies have confirmed a downward frequency shift across blue whale populations [2–6] and a similar trend was noted in fin whales (*Balaenoptera physalus*) [2, 7] and bowhead whales (*Balaena mysticetus*) [8].

The Northeast Pacific population of blue whales is arguably the best studied, and calls from this region have been recorded for decades [9–13]. This population produces three types of low-frequency calls: A and B calls are long-duration (20 s), pulsed and tonal calls, respectively, that often occur in a sequence and are believed to serve a reproductive purpose [14–17] and D calls are downsweeps that last several seconds and are considered a social call, possibly associated with feeding [10, 14, 17]. In the first report on frequency shift in this population, the frequency of the 3$^{rd}$ harmonic of the B call had decreased from 65.7 to 45.5 Hz over 45 years; a rate of 0.4 Hz yr$^{-1}$ from 1963 to 2008 [1].

A variety of hypotheses have been proposed to explain this frequency shift, with explanations ranging from environmental changes to population density and behavioral changes. However, there is currently no consensus on the underlying cause of this frequency decline, as many of the suggested hypotheses, such as an increase in body size, biological masking, and climate change have been refuted [1, 2, 7, 8, 18, 19]. The discovery that this long-term frequency decline was coupled with an intra-annual frequency decline that resets each year, provides further context for the potential mechanism behind this long-term trend [2–4], as does the occurrence of this phenomenon in other baleen whale species [2, 7, 8]. While it is possible that each of these findings have different causes, the most parsimonious explanation would account for both the intra- and inter-annual frequency decline in blue whales, as well as frequency decline in other baleen whale species.

Using passive acoustic recordings collected in recent years, it is possible to (1) examine whether frequency decline is occurring in Northeast Pacific blue whale A calls, (2) examine whether the pulse rate of A calls is changing, and (3) extend the dataset of McDonald *et al.* [1] to observe the current rate of frequency decline in Northeast Pacific blue whale B calls. The results provide insights to further support or refute existing hypotheses and determine if one hypothesis can account for all aspects of this currently unexplained phenomenon.

## Methods

### Data collection

Data were collected from 2006 to 2019 using high-frequency acoustic recording packages (HARPs) deployed at three locations in the Southern California Bight (Fig 1). The three recording locations were sites E (1,300 m), H (1,100 m), and N (1,265 m; Fig 1; S1 Table). HARPs were deployed on the seafloor and had a calibrated hydrophone suspended 10–30 m above the instrument [20]. Data were sampled continuously for all deployments but one, for which the recording was duty cycled (S1 Table). Data were collected at a 200-kHz sampling frequency but were decimated to a 2-kHz sampling frequency to allow more effective scanning of long-term spectral averages (LTSAs). LTSAs were created with 5-s time average and 1-Hz frequency resolution.

### Call measurements

For each year of data from 2006 to 2019, hourly LTSAs were manually scanned during September and October (S1 Table) for high-quality (high signal-to-noise ratio) blue whale A and B

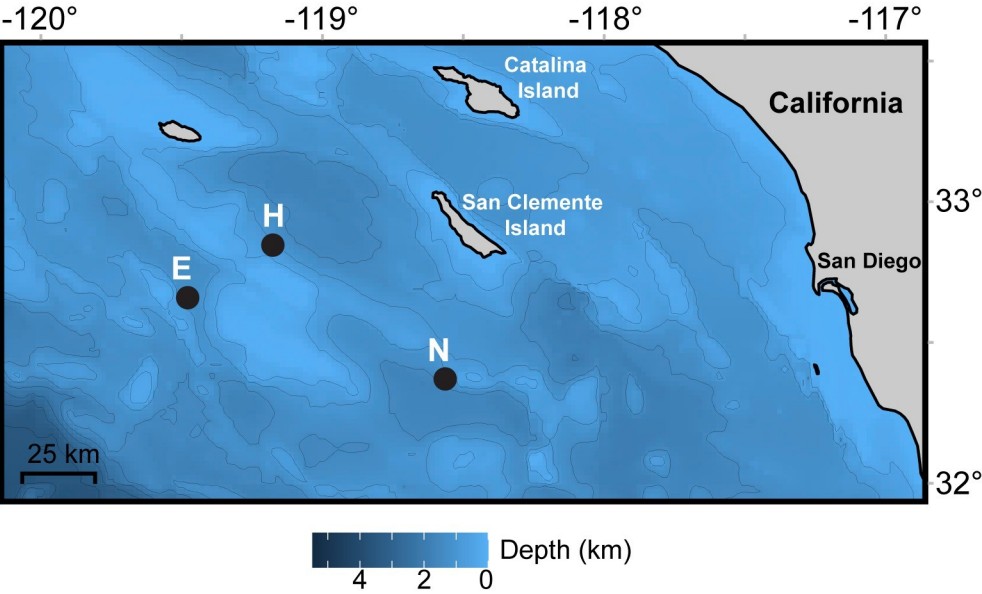

**Fig 1. Map of recording locations offshore of Southern California.** Recording locations H, E, and N are shown as black circles. Color bar indicates depth in kilometers and thin grey lines mark 500 m bathymetry contours. Map generated using the *marmap* and *ggplot2* packages in R [21–23] and bathymetry data from the ETOPO1 database hosted by NOAA [24, 25].

calls using *Triton*, a custom MATLAB program designed to allow visual scans of LTSAs [20]. The following analysis was conducted for A and B calls separately. When a call was identified in the LTSA, a 60-s spectrogram (2,000-point fast Fourier transform (FFT) length, 90% overlap), displayed up to 200 Hz, was used to confirm the quality of the call. When a call was selected for further analysis, custom MATLAB code was used to identify the start and end time of the call. For A calls, average peak frequency was calculated over the full duration of the most dominant call overtone (Fig 2). The number of pulses in each A call was also counted, and it was divided by call duration to determine the pulse rate. For B calls, the peak frequency was measured from the 3rd harmonic of the call every 0.5 s, for the duration of the call. The 3rd harmonic was used, as in McDonald et al. [1], because it typically contains the most energy and is therefore more consistently recorded than the fundamental frequency [10, 12]. The frequency measurement at 10 s from the start of the call was used for subsequent analysis (Fig 2). Before

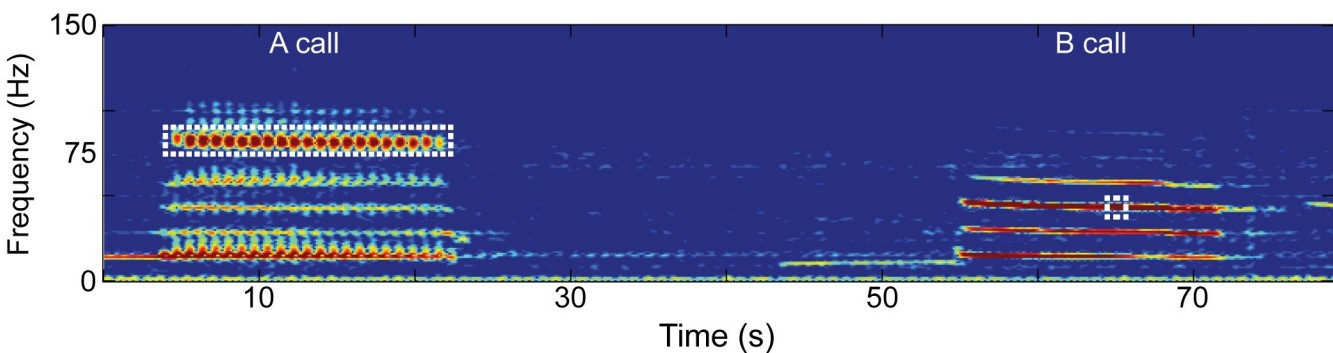

**Fig 2. Northeast Pacific blue whale A and B call.** Calls were recorded on October 17, 2015 at site N. The dashed box indicates the overtone (A call) and harmonic and time point (B call) for which frequencies were used to compare between years.

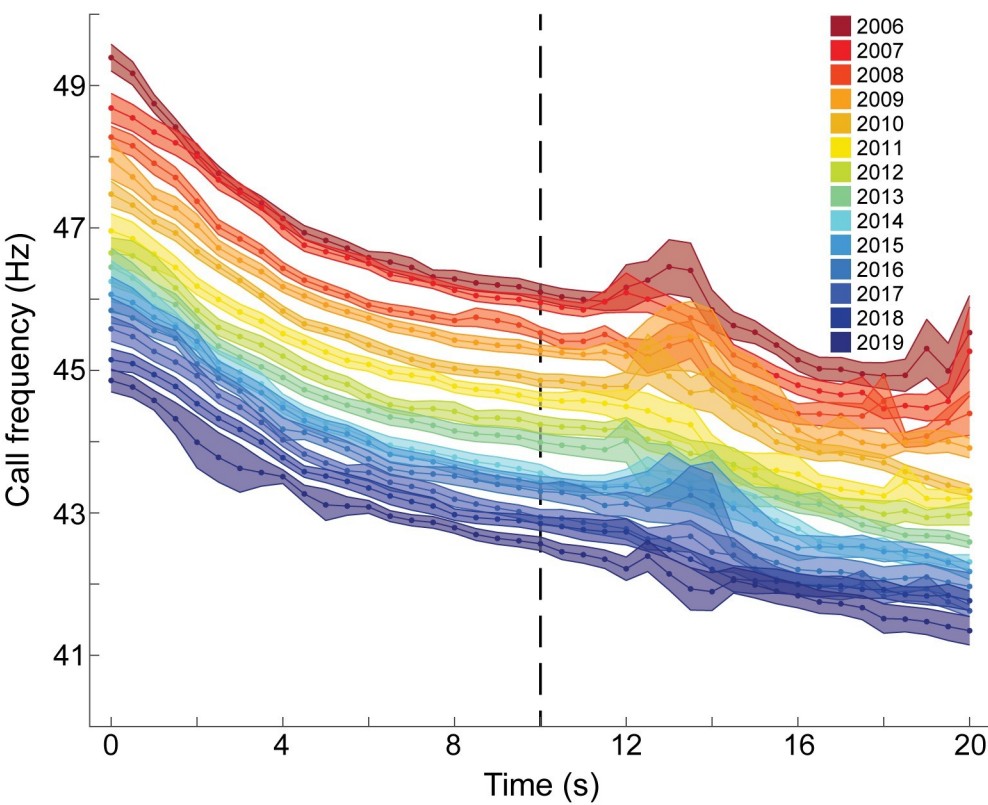

**Fig 3. Mean frequencies of Northeast Pacific blue whale B calls.** The first 20 s of the 3rd harmonic are shown each year from 2006 to 2019. Shaded areas represent 95% confidence intervals and the dashed line indicates 10 s point that was used for yearly comparisons.

B call frequencies were measured, spectrograms were adjusted to an 8,000-point FFT length, resulting in a frequency resolution of 0.25 Hz. Due to the pulsed nature of A calls, a 2,000-point FFT length was used, providing a frequency resolution of 1 Hz.

To select calls for this analysis, we started with visual inspection of data from October 1st and continued until frequency measurements were extracted for 30 calls separated by at least 24 h. If 30 calls were not selected in October, data were scanned starting at the end of September and working backwards until 30 calls had been measured. The months of September and October were selected for frequency measurements because this is a known period of high A and B call production in the Southern California Bight [9, 16, 26]. The sampling routine of 30 calls separated by at least 24 h each was selected to avoid oversampling any individual blue whale.

When selecting the time point to use for comparing B call frequency across years, consideration was given to the methods used by McDonald *et al.* [1], as the goal was to compare the rate of frequency decline in B calls established here to what had been observed previously. For B calls, the frequencies of the 30 calls selected each year were averaged and 95% confidence intervals were calculated along the length of the call (Fig 3). Confidence interval ranges for these yearly averages were lowest from 4.5 to 10 s into the call and highest from 12 to 15 s (Fig 3). The period of high variation is a result of Northeast Pacific blue whale B calls offshore of Southern California exhibiting occasional gaps during this portion of the call [27]. However, only the frequency at 10 s was used for interannual trend analyses. This time point was selected due to the aforementioned low confidence interval ranges and to avoid possible

methodological confounds when comparing results to those reported by McDonald *et al.* [1], who measured frequencies from the estimated midpoint of the call. From the 420 calls measured in this study, the average call duration was 19.6 s, and so we assume that the measurements at 10 s in this study are comparable to those made by McDonald *et al.* [1].

For both A and B calls, the frequencies of the 30 calls measured for a given year were averaged, providing one representative call frequency per year, and 95% confidence intervals were calculated. In the same way, A call pulse rate was also averaged each year. The call measurements from all calls are available in S2 Table and S3 Table, for A and B calls, respectively. To examine interannual trends, the least squares method was used to determine the best fit to the data and the percent of change between years was calculated.

## Results

### A calls

Northeast Pacific blue whale A calls showed a 0.32 Hz $yr^{-1}$ decrease in frequency over 13 years from 2006 to 2019 (Fig 4). The mean frequency was 85.6 Hz in 2006 and 81.5 Hz in 2019 (Fig 4; S1 Table). The best fit for these data was modelled with a second order polynomial ($R^2$ = 0.97). Frequency did not consistently decrease each year; for example, in 2009, 2017, and 2019 the mean frequency was higher than in the previous year (Fig 4). There was no clear trend in the yearly percent of change, which ranged from 0 to 1% (Fig 5).

The A calls showed a 0.006 pulses/s $yr^{-1}$ decrease in pulse rate over the course of the study. The mean pulse rate was 1.241 pulses $s^{-1}$ in 2006 and 1.158 pulses $s^{-1}$ in 2019 (Fig 4; S1 Table). The best fit for pulse rate was modelled with a second order polynomial (R2 = 0.84). However, the pulse rate did not consistently decrease each year, and actually increased in half of the years measured (Fig 4; S1 Table). There was no clear trend in the yearly percent of change, which ranged from -3.3 to 1.3% (Fig 5).

### B calls

Northeast Pacific blue whale B calls showed a 0.27 Hz $yr^{-1}$ decrease in frequency over 13 years from 2006 to 2019 (Fig 6A). The mean frequency at the 10 s point of the 3rd harmonic was 46.1 Hz in 2006 and 42.6 Hz in 2019 (Fig 6; S1 Table). The best fit for this data was modelled with a second order polynomial ($R^2$ = 0.99; Fig 6B).

From 2006 to 2008, our data overlapped with the dataset analyzed by McDonald *et al.* [1] (Fig 6A). The B call frequencies we measured fell within the 95% confidence intervals reported by McDonald *et al.* [1], with a maximum difference in mean frequency of 0.1 Hz over the three years (Fig 6A). The yearly percent of change showed more variation for the frequencies reported by McDonald *et al.* [1] (0–6.6%) compared to those reported here (0.21–1%), and appeared to stabilize after 2000 (Fig 5).

## Discussion

### Frequency and pulse rate decline in Northeast Pacific blue whale calls

From 2006 to 2019, both units of Northeast Pacific blue whale song have been declining, as has been observed in other blue whale populations [18, 19]. We know B calls have been declining in frequency since at least the 1960s [1], and even though there has not previously been a detailed examination of frequency shift in A calls, this decline has been occurring for longer than reported here; descriptions of these calls from offshore of California reveal that in 1997 the frequency of the overtone was around 90 Hz [10]. Although rates of frequency decline

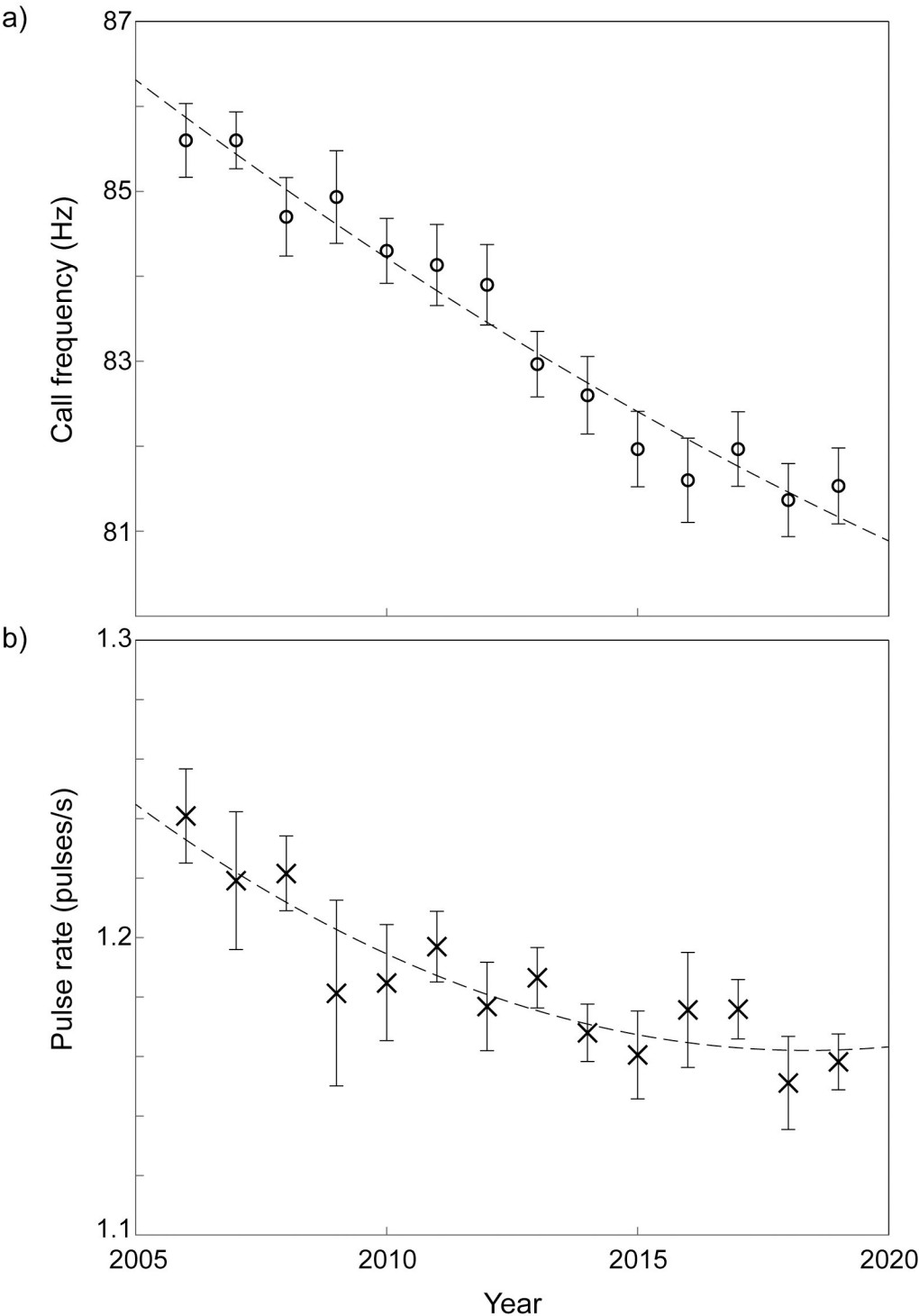

**Fig 4.** Mean frequency (a) and pulse rate (b) of Northeast Pacific blue whale A calls from 2006 to 2019. Error bars represent 95% confidence intervals and the dashed black line represents a quadratic fit to the data.

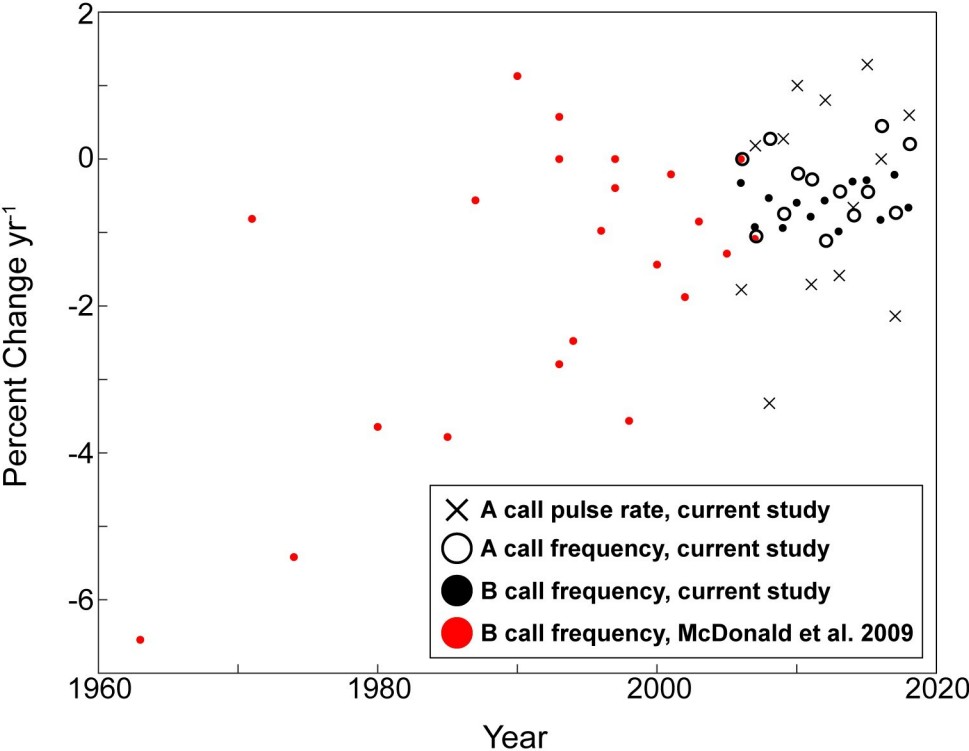

**Fig 5. Percent of change of the mean frequency and pulse rate of Northeast Pacific blue whale calls.** The percent of change is shown for A calls each year from 2006 to 2019 and for B calls from 1963 to 2019. Data in black are from the current study and data in red are from McDonald *et al*. [1].

were comparable between the two calls, the annual change in pulse rate of the A calls did not show the same consistent decline.

For B calls, the rate of decline we report here, at 0.27 Hz yr$^{-1}$, is slower than the rate reported for this population by McDonald *et al*. [1], which was 0.4 Hz yr$^{-1}$ from 1963 to 2008 (Fig 6). These differences are likely not due to methodological inconsistences because, although McDonald *et al*. [1] measured frequency from the "midpoint" of the call and we used the 10-s mark, we found that the frequency of the call showed the least amount of variation from 4.5 to 10 s (Fig 3). Additionally, of the 420 calls that were measured, the average call duration was 19.6 s, suggesting that the call midpoint used by McDonald *et al*. [1] should have been close to the 10-s mark used here. We were also able to show that our measurements fell within the confidence intervals reported by McDonald *et al*. [1] for the years when the two studies overlapped (2006–2008; Fig 6A). Therefore, even if the point at which frequencies were measured was slightly different between studies, it would not explain the differences in rates of decline that we report here.

Another source of variation between our study and McDonald *et al*. [1] is the months that were sampled for calls. Because the frequency of Northeast Pacific blue whale B calls decreases intra-annually as well as inter-annually [4], call frequencies must be measured at the same time each year to accurately evaluate inter-annual trends. To avoid potential confounds from this seasonal influence, we only measured calls recorded in September and October. However, the calls measured by McDonald *et al*. [1] were sampled at various times throughout the years, often whenever data were available. While this may explain some variation with the McDonald *et al*. [1] dataset, during the three years that overlapped with our data, calls were measured in

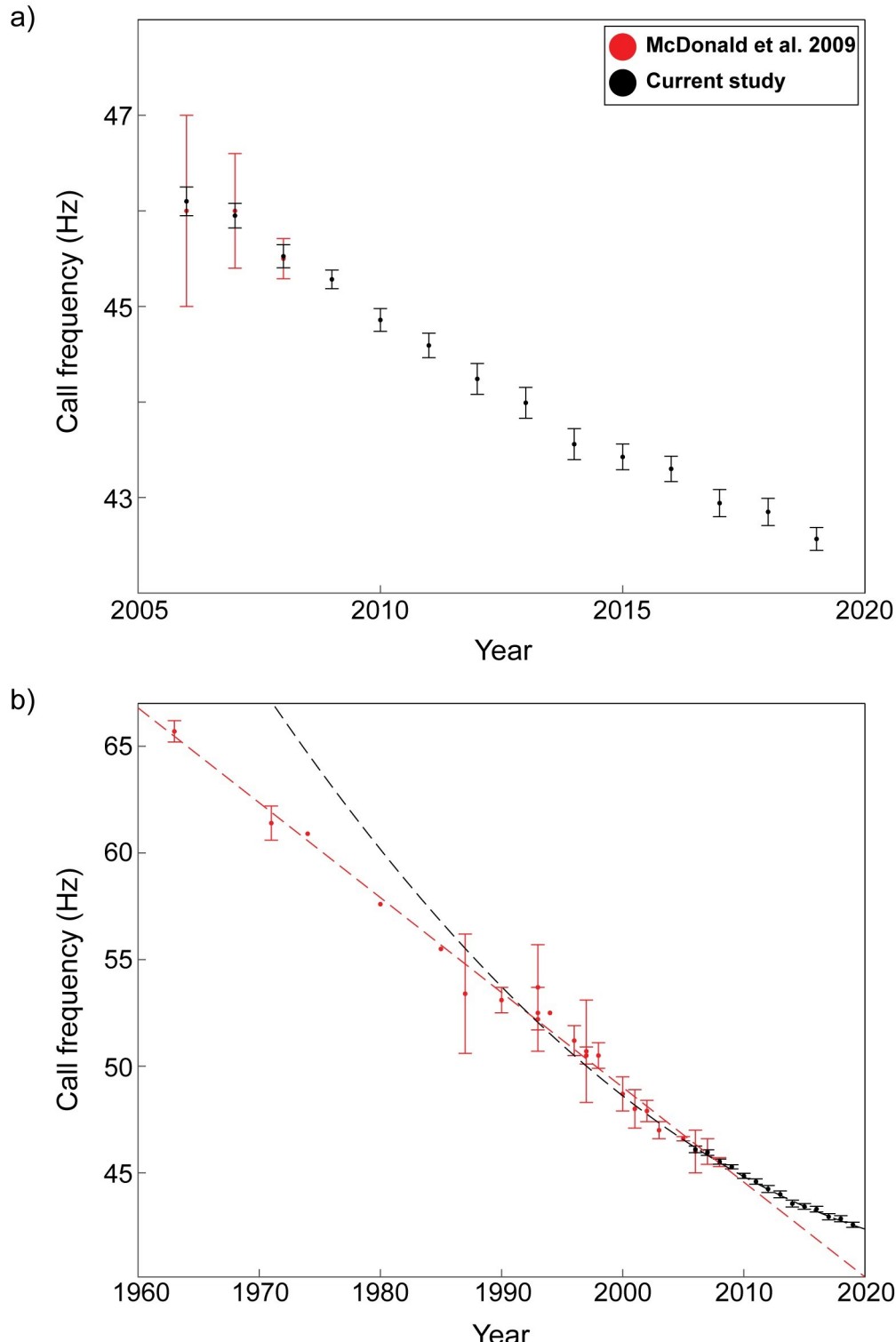

**Fig 6. Mean frequencies of Northeast Pacific blue whale B calls.** Error bars represent 95% confidence intervals for mean frequencies at 10th second of the 3rd harmonic of the calls. Data from McDonald *et al.* [1] are shown in red and data from the current study in black for (a) 2006 to 2019 and (b) 1963 to 2019. The dashed red line represents a linear fit to data from McDonald et al. [1] and the dashed black line represents the quadratic fit to data from the current study.

August and September, and we would not expect a large difference with the frequencies we measured in September and October. Again, this is supported by the comparable frequencies we report for the years where our data overlapped with those of McDonald *et al.* [1]. Therefore, we conclude that rate of frequency decline we report here is not different from that reported by McDonald *et al.* [1] as a result of methodology, and is instead an indication that Northeast Pacific blue whale B calls are declining in frequency at a slower rate than in previous years studied.

## Potential causes of frequency decline

Frequency decline in calls from blue whale populations worldwide [1–6] is evidence of call synchronization within populations, which may provide some adaptive advantages [28]. However, the question remains why this frequency decline is occurring and whether the new information presented, that A call frequency and pulse rate are also decreasing and that B call frequency is declining at a slower rate than reported previously, point towards one of the previously proposed hypotheses regarding this phenomenon. The decrease in the rate of decline may be a simple result of a physical limit being reached for vocalization frequency, but it is worth considering whether there is an alternative explanation for this observation. For the previously proposed hypotheses, we must consider whether they can explain all aspects of this phenomenon: the inter- and intra-annual frequency decline in blue whales worldwide, the decrease in the rate of frequency decline observed here in blue whales, and the observed frequency decline in other baleen whale species. While it is possible that multiple factors are responsible for these different observations, the most parsimonious explanation would provide a root cause that accounts for all these observations.

Hypotheses on seasonal changes in body condition [29] and calling depth [5] have been suggested as explanations for intra-annual frequency decline, but it is understood that these theories cannot explain the inter-annual trend [3, 29]. For calling depth, although call frequency can be impacted by depth, there is no biological basis to explain why blue whales, and other species for which frequency shift has been documented, would all collectively be calling at greater depths each year. Tag data also shows that calling depth can vary over a short time period [17], but because call frequency shows limited variation [1, 3, 18, 19, 28], it does not appear that call frequency is solely influenced by depth [19]. Therefore, we will not examine these hypotheses in further detail.

Although we know that ocean ambient noise has increased over the period that blue whale calls have declined in frequency [30–32], the expected response to avoidance of masking would be to call at higher frequencies, as a result of producing higher amplitude calls, as has been observed in right whales [33, 34]. Therefore, if blue whales were trying to be heard in a louder ocean, we would expect their calls to be increasing in frequency, not decreasing. Additionally, ambient noise has actually decreased in the Southern Indian Ocean, but blue whale calls in this region still exhibit a frequency decline [2], meaning that ambient noise does not provide a consistent explanation for changing blue whale call frequencies. However, Leroy *et al.* [2] suggested that seasonal changes in ambient noise could still explain intra-annual frequency decline because, at multiple sites in the Southern Indian Ocean, intra-annual frequency change in both Antarctic blue whales and fin whales was correlated with seasonal changes in ambient noise levels [2]. While it is possible that different factors are driving intra- and inter-annual frequency decline, the evidence presented is only that of two seasonal patterns. Blue whales are already known to exhibit seasonal changes in their acoustic behavior [16]; therefore, annual fluctuations in ambient noise should not necessarily be accepted as the driver of intra-annual blue whale frequency decline.

One theory originally proposed by McDonald *et al.* [1] was that call frequency is related to population density. This theory suggests that blue whales called at higher amplitudes, and consequently higher frequencies, to be heard by conspecifics after their populations were reduced by commercial whaling. As populations have increased, blue whales have been able to call at lower amplitudes because the distance between conspecifics has decreased. This assumes that there is a trade-off between frequency and amplitude that males exploit to produce lower frequency calls that may serve as an indicator of body size and fitness in inter- or intra-sexual selection [1]. Therefore, this theory could conceivably explain inter-annual frequency decline in other baleen whale species that were also impacted by commercial whaling, as well as account for the differing rates of frequency decline observed in different populations, as each population presumably recovers at its own rate [18]. However, offshore of Australia, inter- and intra-annual frequency decline has been documented from 2002 to 2017 in the spot call, which is presumed to be from a baleen whale, though the species has yet to be confirmed [35]. This call underwent a dramatic (>5 Hz) increase in frequency between 2006 and 2007 and then began to gradually decline again through 2017 [35]. This abrupt frequency increase could not be explained by the population density theory. Additionally, McDonald *et al.* [1] hypothesized that, if this theory were the mechanism behind blue whale frequency shift, song frequencies would stabilize in conjunction with population densities. The Northeast Pacific blue whale population was reduced by historical whaling [36, 37] and is still considered endangered. Although there is evidence the population may have returned to pre-whaling levels [38], recent data suggest that this population has experienced a recent growth after relatively stable numbers between the mid-1990s and mid-2010s [39]. It is not clear whether the change in the rate of decline we observed matches these variations in abundance trends.

The population density theory also assumes a relationship between source level and frequency that has not been observed in blue whales [5] and has been refuted in bowhead whales [8]. Finally, this theory does not account for intra-annual frequency decline, as it would require that animal densities steadily change within each year. Off the U.S. West Coast, blue whale distribution is not consistent between years, likely related to shifts in oceanographic conditions [40–42]. In bowhead whales, Thode *et al.* [8] found that a decrease in call frequency was predicted by an increase in call density, but this relationship has not been examined in blue whales. Therefore, the population density theory may not serve as a parsimonious explanation for all aspects of the frequency-shift phenomenon.

Finally, the decrease in pulse rate, documented here in A calls, has been previously observed in other blue whale populations' songs [6, 19]. Changes in pulse rate have also been documented in fin whale populations [7, 43], though these are inter-call rates as opposed to the intra-call rates described for blue whales. These observations add an additional layer of complexity to the frequency shift phenomenon and are not explained by any of the previously discussed theories.

Because A calls are a pulsed signal and frequency shift was measured from an overtone, it is possible that the amplitude modulation of the signal was changing and causing the documented frequency shift of the overtone, rather than it being caused by the shift in the carrier frequency. However, although pulse rate declined over the years examined, the decline was not consistent each year and did not directly correspond to the frequency decline of the same year. For example, from 2008 to 2009 the frequency increased slightly but the pulse rate decreased, whereas from 2010 to 2011 the frequency decreased and the pulse rate increased. If future studies are to examine frequency decline in A calls, it might be worthwhile to measure frequencies of both carrier frequency and the most energetic overtone to be able to eliminate the possibility of frequency shift being caused by amplitude modulation changes, and to compare to the frequencies reported here. It might also be valuable to examine the received levels of the

measured calls, as it is possible that variations in propagation and thus received level could have affected measured frequencies.

Much of the discussion around frequency shift has been focused on the production aspects and possible mechanisms and limits in sound production. However, it is also worth considering the role perception may play in baleen whales, especially for small frequency changes. Baleen whale auditory perception is poorly understood, although we know they have a thick auditory cortex [44] and we can assume that, as in other species, auditory processing depends on auditory as well as behavioral stimuli [45]. One of the unresolved questions is, can the whales perceive this gradual frequency shift? We do not know the frequency difference limen, the smallest detectable change in frequency, for baleen whales, but if we assume it is similar to humans', their ability to distinguish small frequency changes could range between 0.2 and 0.09 Hz at 45 Hz [46]. At the high end, it would mean that whales might be able to perceive the overall seasonal change, but their consistent smaller frequency adjustment throughout the season [3] could be below the whale's perceptive abilities of frequency differences. If the whales cannot, in fact, perceive this change, the gradual frequency shift may be an unintended consequence of a currently not understood song synchronization mechanism. It is clear we still have much to learn about whale auditory perception, but the ubiquitous nature of the frequency shift phenomenon may indicate a consistent level of vocal plasticity and fine auditory processing abilities across baleen whale species.

## Conclusions

Examination of Northeast Pacific blue whale song units from 2008 to 2019 revealed a frequency decline in A calls, as well as a decrease in the rate of frequency decline in B calls from that reported for the second half of the 20th century. A decrease in the pulse rate of A calls was also documented. With this new information, we were able to revisit previously proposed hypotheses that have attempted to explain the worldwide frequency decline of blue whale song [1]. Although none of the currently proposed theories successfully account for all aspects of the frequency shift phenomenon, documentation of frequency decline and other changes to call characteristics in baleen whale species should continue in an effort to discover the mechanism underlying this trend.

## Supporting information

**S1 Table. High-frequency acoustic recording package deployment details.** The year, site, latitude, longitude, depth, and analysis period for each recording package deployment are provided, along with the associated mean frequency (and pulse rate for A calls) and 95% confidence intervals (CI) for both A and B calls each year.
(PDF)

**S2 Table. Northeast Pacific blue whale A call measurements.** Call duration, average peak frequency, number of pulses, and pulse rate for each call measured each year from 2006 to 2019.
(XLSX)

**S3 Table. Northeast Pacific blue whale B call measurements.** Frequency measurements of the 3rd harmonic for each call measured each year from 2006 to 2019. NaNs signify that the call did not extend to the specified time point.
(XLSX)

## Acknowledgments

We thank Bruce Thayre, John Hurwitz, Ryan Griswold, Sean Wiggins, Rohen Gresalfi, Jenny Trickey, Brent Hurley, Chris Garsha, and Erin O'Neill for coordinating instrument deployment, recovery, and data processing.

## Author Contributions

**Conceptualization:** Ana Širović, John A. Hildebrand.

**Formal analysis:** Ally Rice, Megan Wood, Alex Carbaugh-Rutland.

**Funding acquisition:** Ana Širović, John A. Hildebrand, Simone Baumann-Pickering.

**Methodology:** Ally Rice, Ana Širović, John A. Hildebrand.

**Supervision:** Ana Širović, John A. Hildebrand, Simone Baumann-Pickering.

**Visualization:** Ally Rice.

**Writing – original draft:** Ally Rice.

**Writing – review & editing:** Ally Rice, Ana Širović, John A. Hildebrand, Megan Wood, Alex Carbaugh-Rutland, Simone Baumann-Pickering.

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
