## [Decision Letter · Decision Letter 0]

2 Dec 2021

PONE-D-21-29018Update on frequency decline of Northeast Pacific blue whale (Balaenoptera musculus) callsPLOS ONE

Dear Dr. Rice,

Thank you for submitting your manuscript to PLOS ONE. After careful consideration, we feel that it has merit but does not fully meet PLOS ONE’s publication criteria as it currently stands. Therefore, we invite you to submit a revised version of the manuscript that addresses the points raised during the review process. This manuscript has now been assessed by two experts. Both were quite positive about the study, but have a number of recommendations for improving it. Please thoroughly address all reviewer comments when revising your manuscript. Please submit your revised manuscript by Jan 16 2022 11:59PM. If you will need more time than this to complete your revisions, please reply to this message or contact the journal office at plosone@plos.org. Please include the following items when submitting your revised manuscript:A rebuttal letter that responds to each point raised by the academic editor and reviewer(s). You should upload this letter as a separate file labeled 'Response to Reviewers'.A marked-up copy of your manuscript that highlights changes made to the original version. You should upload this as a separate file labeled 'Revised Manuscript with Track Changes'.An unmarked version of your revised paper without tracked changes. You should upload this as a separate file labeled 'Manuscript'.

We look forward to receiving your revised manuscript.

Kind regards,

William David Halliday, Ph.D.

Academic Editor

PLOS ONE

Journal Requirements:

2. We note that Figure 1 in your submission contain map images which may be copyrighted. All PLOS content is published under the Creative Commons Attribution License (CC BY 4.0), which means that the manuscript, images, and Supporting Information files will be freely available online, and any third party is permitted to access, download, copy, distribute, and use these materials in any way, even commercially, with proper attribution. For these reasons, we cannot publish previously copyrighted maps or satellite images created using proprietary data, such as Google software (Google Maps, Street View, and Earth). For more information, see our copyright guidelines: http://journals.plos.org/plosone/s/licenses-and-copyright.

 a. You may seek permission from the original copyright holder of Figure(s) [#] to publish the content specifically under the CC BY 4.0 license. 

Reviewers' comments:

Reviewer's Responses to Questions

**Comments to the Author**

1. Is the manuscript technically sound, and do the data support the conclusions?

Reviewer #1: Yes

Reviewer #2: Partly

2. Has the statistical analysis been performed appropriately and rigorously? 

Reviewer #1: Yes

Reviewer #2: No

3. Have the authors made all data underlying the findings in their manuscript fully available?

Reviewer #1: Yes

Reviewer #2: Yes

4. Is the manuscript presented in an intelligible fashion and written in standard English?

Reviewer #1: Yes

Reviewer #2: Yes

5. Review Comments to the Author

Reviewer #1: Review of Frequency decline in blue whale calls

Firstly, my apologies to the authors for the delay in providing the review. Just too heavy a a workload.

The paper "Review of Frequency decline in blue whale calls" is well written and requires no grammatical edits. My suggestions are to do with: 1) the interpretation of frequency shifts between fundamental frequencies and harmonics or overtones; and 2) some additional ideas for discussion.

The authors have used techniques widely used by others. There is the potential for errors of interpretation in what is being observed (which applies to earlier works also) so I am suggesting some additional analysis as per below.

Main comment:

1/

Harmonics are caused by amplitude modulation of a carrier tone. The harmonic spacing relates to the frequency of amplitude modulation. If there is no amplitude modulation of the carrier tone then there are no harmonics or overtones. Thus if you measure frequency shift of harmonics there are potentially two causes which can act independently: 1) a change in frequency of the carrier tone which would result in a linear change in frequency of the harmonic, assuming the harmonic amplitude modulation rate did not change (this is what your analysis and those of others assumes); or 2) a change in the amplitude modulation rate which would result in the overtone or harmonic having a different frequency 'slope' through a call than the carrier frequency. Thus if you measure a frequency shift of the harmonic you need to be confident that the amplitude modulation rate which produces the harmonic has stayed constant through the call, if you are to assume the frequency shift applies to the entire call.

At a worst case it is possible the whale call carrier frequency has not changed but the amplitude modulation rate has, which will be reflected in a frequency decline of the harmonics. While this is not the case with great whales and the call carrier frequency does decline over time there needs to be some better rigour applied when changes in frequencies of harmonics are analysed and in particular compared between authors, whom for the same species may be comparing frequency changes measured between harmonics and carrier frequencies, which are generated by different phenomena (carrier frequency by the physics of the sound generation apparatus and harmonics by the amplitude modulation rate of the carrier frequency).

If you are to compare frequency rates with the earlier study of McDonald et al, then you should be comparing the same harmonics.

While it means more analysis, it would also be good (and provide more rigour) if you could compare rates of change of carrier frequency with harmonic frequency (a reflection of amplitude modulation rate) through a call to determine which is changing - carrier frequency, amplitude modulation rate, or both? If the amplitude modulation rate does not change within a call then your analysis (and all the other published studies which use harmonic frequencies to demonstrate frequency changes over time) stands as they are. If the amplitude modulation changes through time then that opens new questions on what is going on and confuses the yearly comparisons, particularly the amount of frequency decline observed in different studies.

I do think you should do this analysis, otherwise we perpetuate the potential (am not saying it is always present or has occurred here) confusion of what is being measured - changes to carrier frequency, amplitude modulation rate if harmonics are measured, or both?

It would also help if you had a physicist to elaborate in the introduction, the production of harmonics or overtones and how this impacts what you are trying to measure (frequency shifts across a long time frame).

2/

Extra points for discussion:

Ward et al. (2017) show a 'resetting' of carrier frequency (this call type has no overtones) in a great whale species (see Fig. 5). This species should be 'resetting' its carrier frequency again, around now. It may be worth mentioning this. The 'resetting' implies a conscious effort on the part of the animal to shift its frequency as it occurs over such a short time period (one or two seasons) which is far too short to represent an environmental shift.

I disagree with the impact depth can have on whale carrier frequency. You have implied that depth has little impact. Tags attached to whales often have poor resolution (0.5 m is common), thus require high sample rates to infer small scale trends of changes in depth (ie. 20 s across a call length). We have a work in progress using a high sample rate tag which suggests call frequency within a blue whale species' call is related to the whale's depth. This would imply the animal can consciously change its call carrier frequency by changing its singing depth by a small amount. The species in question does this within a call. You can ignore this if you wish but I would suggest you at least say the animals singing depth will play a role in call carrier frequency changes.

Ward, R., Gavrilov, A. N., McCauley, R.D. (2017) “Spot” call: a common sound from an unidentified great whale in Australian temperate waters. J. Acoust. Soc. Am. Express Letters 142(2) EL231-236, doi.org/10.1121/1.4998608

Reviewer #2: Please see attachment for comments and plot of pulse rate vs. time.

Basic comments are that statistical regression needs to control for other factors, and analysis should also include pulse rate for type A calls.

6. PLOS authors have the option to publish the peer review history of their article (what does this mean?). If published, this will include your full peer review and any attached files.

Reviewer #1: **Yes: **Robert McCauley

Reviewer #2: **Yes: **Aaron Thode

---

## [Author Response · Author response to Decision Letter 0]

14 Jan 2022

Journal Requirements: There was a copyright concern over the map used in Figure 1. However, the bathymetry data used in this map is from the ETOPO1 database that is publicly available from NOAA, so there are no copyright concerns with the use of this image. However, for clarity, we have added the source of the bathymetry data to the figure caption and include appropriate references for the use of this data, as requested by NOAA. 

Reviewer 1 Responses:

Comment 1: Harmonics are caused by amplitude modulation of a carrier tone. The harmonic spacing relates to the frequency of amplitude modulation. If there is no amplitude modulation of the carrier tone then there are no harmonics or overtones. Thus if you measure frequency shift of harmonics there are potentially two causes which can act independently: 1) a change in frequency of the carrier tone which would result in a linear change in frequency of the harmonic, assuming the harmonic amplitude modulation rate did not change (this is what your analysis and those of others assumes); or 2) a change in the amplitude modulation rate which would result in the overtone or harmonic having a different frequency 'slope' through a call than the carrier frequency. Thus if you measure a frequency shift of the harmonic you need to be confident that the amplitude modulation rate which produces the harmonic has stayed constant through the call, if you are to assume the frequency shift applies to the entire call.

At a worst case it is possible the whale call carrier frequency has not changed but the amplitude modulation rate has, which will be reflected in a frequency decline of the harmonics. While this is not the case with great whales and the call carrier frequency does decline over time there needs to be some better rigour applied when changes in frequencies of harmonics are analysed and in particular compared between authors, whom for the same species may be comparing frequency changes measured between harmonics and carrier frequencies, which are generated by different phenomena (carrier frequency by the physics of the sound generation apparatus and harmonics by the amplitude modulation rate of the carrier frequency).

If you are to compare frequency rates with the earlier study of McDonald et al, then you should be comparing the same harmonics.

While it means more analysis, it would also be good (and provide more rigour) if you could compare rates of change of carrier frequency with harmonic frequency (a reflection of amplitude modulation rate) through a call to determine which is changing - carrier frequency, amplitude modulation rate, or both? If the amplitude modulation rate does not change within a call then your analysis (and all the other published studies which use harmonic frequencies to demonstrate frequency changes over time) stands as they are. If the amplitude modulation changes through time then that opens new questions on what is going on and confuses the yearly comparisons, particularly the amount of frequency decline observed in different studies.

I do think you should do this analysis, otherwise we perpetuate the potential (am not saying it is always present or has occurred here) confusion of what is being measured - changes to carrier frequency, amplitude modulation rate if harmonics are measured, or both?

It would also help if you had a physicist to elaborate in the introduction, the production of harmonics or overtones and how this impacts what you are trying to measure (frequency shifts across a long time frame).

Response: True harmonics, integer multiples of the fundamental frequency, are not caused by amplitude modulation of a carrier tone, as commented, but result from distortion of the signal wave in the vocal tract. The B calls of Northeast Pacific blue whale songs are not amplitude modulated signals and have true harmonics. An amplitude modulated signal, such as a Northeast Pacific blue whale A call, has non-harmonic overtones (or side bands). Therefore, measuring the 3rd harmonic of blue whale B calls is an accurate way to measure frequency shift, and is comparable across studies, regardless of which harmonic is being measured (provided the harmonic multiplier is accounted). Nevertheless, to demonstrate this, we used a subset of our data (20 calls, every other year) and performed the same analysis as described in our study but using the fundamental frequency of the B calls. We found that the fundamental frequency was declining at a rate of 0.09 Hz/yr, which corresponds to the 0.27 Hz/yr decline we report for the 3rd harmonic. In the figure below, which shows the average frequency measurements and 95% confidence intervals for the first 20 s of the calls for each year, you can see that the confidence intervals are larger than reported for the 3rd harmonic (shown in the manuscript in Figure 3) because the fundamental frequency is often faint and can be masked by other low frequency signals, such as from fin whale 20 Hz calls. As stated in the manuscript, the 3rd harmonic typically contains the most energy and can therefore be more consistently and accurately measured than the fundamental. Therefore, we believe that frequency shift should be measured using the 3rd harmonic instead of the fundamental.

Although we conducted this analysis, we feel it is not worth including in the manuscript, as it is simply a confirmation of what we would expect based on the physics behind the production of harmonics. Additionally, the McDonald et al. (2009) study that we compare our rate of frequency decline to was also measuring the 3rd harmonic. This was stated in the manuscript introduction but is also now restated in the methods, to avoid potential confusion (lines 98–99).

However, because blue whale A calls are amplitude modulated signals, and we were measuring frequency shift using an overtone for this call type, it is possible that amplitude modulation had some influence on its frequency decline. Reviewer 2 commented that there was a decline in the pulse rate of A calls over the years of the study, which was not something we investigated in depth. We have now included an examination of the decline in A call pulse rate in the manuscript as suggested, which includes discussion of the relationship between carrier frequency and overtones (lines 290–302). So, while it is certainly true that it will be important in future studies to take this into consideration when measuring frequency shift in blue whale A calls, we currently do not have the amplitude measurements in hand. We suggest that our proposed edits sufficiently address the reviewer concerns.

Comment 2: Extra points for discussion:

Ward et al. (2017) show a 'resetting' of carrier frequency (this call type has no overtones) in a great whale species (see Fig. 5). This species should be 'resetting' its carrier frequency again, around now. It may be worth mentioning this. The 'resetting' implies a conscious effort on the part of the animal to shift its frequency as it occurs over such a short time period (one or two seasons) which is far too short to represent an environmental shift.

I disagree with the impact depth can have on whale carrier frequency. You have implied that depth has little impact. Tags attached to whales often have poor resolution (0.5 m is common), thus require high sample rates to infer small scale trends of changes in depth (ie. 20 s across a call length). We have a work in progress using a high sample rate tag which suggests call frequency within a blue whale species' call is related to the whale's depth. This would imply the animal can consciously change its call carrier frequency by changing its singing depth by a small amount. The species in question does this within a call. You can ignore this if you wish but I would suggest you at least say the animals singing depth will play a role in call carrier frequency changes.

Response: We have added a brief discussion of the Ward et al. (2017) paper where we describe evidence to refute the abundance hypothesis (lines 262�267).

We did not mean to imply that depth has little impact on call frequency, only that the impact it could have does not adequately explain the inter-annual frequency shift. If a decrease in call depth were the explanation behind the frequency shift phenomenon it would mean that blue whales are collectively calling at greater depth each year. Without evidence to support this, or a biological precedent upon which to base such a claim, we did not want to speculate on this further. The evidence we have against this explanation is the fact that tag data has shown that calling depth can change over a short time period (Lewis et al. 2018). If frequency was progressively changing with depth, we would expect call frequencies to be much more variable within a year than they are reported to be. So, although depth may be a factor in frequency changes in some calls and at some times, it does not adequately explain the consistent frequency shift phenomenon based on the information currently available. We have made this assertion clearer in the text (lines 229�234).

Reviewer 2 Responses:

Comment 1: Statistical regression: The paper performs a simple least-squares fit to the frequency data of both types of calls. Unfortunately, apparent peak frequency can be affected by factors other than the source (such as range and instrumentation), and the dataset is large enough that some of these factors can be checked. At a minimum the authors should use a generalized linear model (GLM) using received level as a predictor variable, HARP unit/location as a categorical predictor variable, and year as another predictor variable. The importance of received level is that it is a proxy for horizontal range, and there is abundant evidence that range and environmental propagation conditions impact peak frequency in other animals. Even if the water depths here seem deep (1 km in some locations), frequency-dependent attenuation and bottom interactions cannot be ruled out. 

In addition, the paper should clarify whether the FFT length used to analyze the peak frequency of the type A call was shorter than the pulse rate. If the FFT length is too long, then the spectrum of the type A call will shift from the structure shown in Figure 2 to a “comb-like” spectrum where the separation between comb peaks is the reciprocal of the pulse rate. It’s important to ensure that the 75 Hz frequency peak measured is due to the “formant” shift and not from changes in the pulse rate. 

Response: The influence of received level is not a concern for B calls because we were comparing frequencies that were measured at a point in the call when the signal becomes nearly tonal. Because these calls are essentially a tone and not a broadband signal, there would be no change to the peak frequency we measured as a result of propagation. Even if our measurements varied by a few seconds within the call, the frequency change would be so small (<0.5 Hz) that it would not be sufficiently impacted by propagation but would contribute to the relatively tight confidence intervals. 

However, this is a potential concern for A calls, for which frequency measurements were averaged over the duration of the call. Unfortunately, determining the received level of each call would require substantial additional re-analysis because we did not extract received level information with our initial measurements and this re-analysis was not feasible at this time. Although we were not able to include the GLM as requested, we have included a more thorough discussion of changes in A calls, since we now incorporate an analysis of pulse rate changes, and used that to also comment on this concern (lines 284–302). 

The FFT length used for the A call analysis was 2000 (sampling frequency was 2000 Hz, resulting in a 1-Hz resolution), as stated in the methods, and is therefore shorter than the pulse rate. There were 2 calls measured (out of 420) for which pulse rate was 1 s, and one where it was 0.9 s, but all other calls had a pulse rate above 1 s. This is certainly a factor that should be considered if future studies were to compare measurements of A calls to ours. 

Comment 2: Pulse rate analysis: the attached figure, generated from supplemental table S2, shows that the pulse rate of type A calls is decreasing over time, to the point that by 2019 the rate has shifted one standard deviation from 2006. This would suggest that a more rigorous regression would find a significant relationship between pulse rate and time, an observation that may have a bearing on the extensive discussion. 

Response: We now include an analysis of A call pulse rate in the manuscript. Thank you for this suggestion.

Comment 3: About that extensive discussion: I would make it shorter, as a detailed list of speculations does not enhance the paper quality. 

Response: We removed one paragraph from the discussion that described 4 theories that have been refuted in all other papers on this subject. These are now very briefly referenced in the introduction (lines 57–59). For the other theories, we feel they are still worth discussing as previous publication have argued that they may explain some part of the frequency decline phenomenon, while we argue for a more parsimonious explanation.

---

## [Decision Letter · Decision Letter 1]

22 Mar 2022

Update on frequency decline of Northeast Pacific blue whale (Balaenoptera musculus) calls

PONE-D-21-29018R1

Dear Dr. Rice,

We’re pleased to inform you that your manuscript has been judged scientifically suitable for publication and will be formally accepted for publication once it meets all outstanding technical requirements. Both reviewers assessed your revisions, and feel that you have addressed all of their comments appropriately.

Kind regards,

William David Halliday, Ph.D.

Academic Editor

PLOS ONE

Additional Editor Comments (optional):

Reviewers' comments:

Reviewer's Responses to Questions

**Comments to the Author**

1. If the authors have adequately addressed your comments raised in a previous round of review and you feel that this manuscript is now acceptable for publication, you may indicate that here to bypass the “Comments to the Author” section, enter your conflict of interest statement in the “Confidential to Editor” section, and submit your "Accept" recommendation.

Reviewer #1: All comments have been addressed

2. Is the manuscript technically sound, and do the data support the conclusions?

Reviewer #1: Yes

3. Has the statistical analysis been performed appropriately and rigorously? 

Reviewer #1: Yes

4. Have the authors made all data underlying the findings in their manuscript fully available?

Reviewer #1: Yes

5. Is the manuscript presented in an intelligible fashion and written in standard English?

Reviewer #1: Yes

6. Review Comments to the Author

Reviewer #1: Thank you to the paper authors for considering and addressing comments. A comment, to decrease call frequency for a bubble the bubble depth must decrease, only by a small amount for the frequency shifts observed. This would apply to a whale lung space. Perhaps there are differing mechanisms for producing frequency shifts amongst different whale species, which would make the long term frequency decline more puzzling.

7. PLOS authors have the option to publish the peer review history of their article (what does this mean?). If published, this will include your full peer review and any attached files.

Reviewer #1: No

---

## [Editor Report · Acceptance letter]

24 Mar 2022

PONE-D-21-29018R1 

Update on frequency decline of Northeast Pacific blue whale (*Balaenoptera musculus*) calls 

Dear Dr. Rice:

I'm pleased to inform you that your manuscript has been deemed suitable for publication in PLOS ONE. Congratulations! Your manuscript is now with our production department. 

Kind regards, 

on behalf of

Dr. William David Halliday 

Academic Editor

PLOS ONE